# Creutzfeldt–Jakob Disease with a Five-Year Clinical Course, Multicentric Cerebellar Prion Plaques and Prior History of Biopsy-Proven Primary Angiitis of the Central Nervous System: A Case for Iatrogenic Exposure?

**DOI:** 10.3390/v12121411

**Published:** 2020-12-08

**Authors:** Kristina Jeon, Jeffrey T. Joseph, Gerard H. Jansen, Anne Peterson, J. David Knox, Valerie L. Sim

**Affiliations:** 1Department of Medicine, University of Alberta, Edmonton, AB T6G 2R3, Canada; kjeon@ualberta.ca; 2Department of Pathology, University of Calgary, Calgary, AB T2N 1N4, Canada; jtjoseph@ucalgary.ca; 3Department of Pathology and Laboratory Medicine, University of Ottawa, Ottawa, ON K1N 6N5, Canada; gjansen@eorla.ca; 4National Microbiology Laboratory, Public Health Agency of Canada, Winnipeg, MB R3E 3R2, Canada; anne.peterson@canada.ca (A.P.); david.knox2@canada.ca (J.D.K.); 5Centre for Prions and Protein Folding Diseases, University of Alberta, Edmonton, AB T6G 2R3, Canada

**Keywords:** Creutzfeldt-Jakob disease, primary angiitis of the central nervous system, iatrogenic, prion, pathology, multicentric plaques, primary progressive aphasia, apraxia

## Abstract

Creutzfeldt–Jakob disease (CJD) is a rapidly progressive neurodegenerative disease that can arise spontaneously, genetically, or be acquired through iatrogenic exposure. Most patients die within a year of symptom onset. It is rare, affecting 1–2 per million per year, and the majority of cases are sporadic. Primary angiitis of the central nervous system (PACNS) is also rare, affecting 2.4 per million per year. We present a case of an unusually long clinical course of CJD, almost five years, which began with symptoms of apraxia. The patient had biopsy-proven PACNS 16 years prior to clinical presentation, and the site of biopsy was the left parietal lobe. Autopsy revealed multicentric prion plaques in the cerebellum, in the setting of normal genetic testing. The presence of plaques in the cerebellum, and prior neurosurgery, raises the possibility of iatrogenic exposure. We present the details of this case, including pathology from the original biopsy and final autopsy, as well as a review of relevant cases in the literature.

## 1. Introduction

Prion diseases are categorized into three groups: sporadic, genetic, and acquired (variant or iatrogenic), comprising 85–90%, 10–15%, and less than 1% of cases, respectively [1]. Sporadic Creutzfeldt–Jakob disease (sCJD) is the most common type of prion disease that affects humans, while still being a rare disease with one new case per 1,000,000 individuals each year. Genetic prion diseases, such as genetic CJD (gCJD), fatal familial insomnia (FFI), and Gerstmann–Sträussler–Scheinker syndrome (GSS), involve several different prion protein gene (*PRNP*) mutations. Unlike sCJD, iatrogenic CJD (iCJD) is often associated with a known origin of the disease, historically with cadaveric pituitary hormones, dural graft transplants, and corneal transplants. The incubation period for iCJD can vary from 1 to 42 years, with a mean incubation period of 9–10 years [2,3].

CJD is pathologically characterized by neuronal loss, proliferation of glial cells, presence of spongiform change within the neuropil, and the presence of protease-resistant prion protein (PrP^Sc^) [1]. Disease onset is often preceded by long asymptomatic incubation periods followed by rapid deterioration. The most common presenting symptoms and signs include dementia, ataxia, behavioral abnormalities, and higher cortical dysfunction (aphasia, apraxia, and frontal lobe syndromes). They may also present with startle myoclonus, nystagmus and ataxia, hyperreflexia, Babinski reflex, spasticity, and extrapyramidal signs [1]. After symptom onset, disease duration can range vastly, but is usually between 4 and 18 months, depending on the subtype of CJD [4]. All types of CJD are universally fatal.

Primary angiitis of the central nervous system (PACNS) is unlike CJD in that the pathogenesis of PACNS is dependent on inflammation of the small- and medium-sized arteries of the CNS. PACNS is also a rare disease, with a reported annual incidence rate of 2.4 cases per 1,000,000 person-years [5]. This inflammation results in CNS dysfunction, which can present as headache, dementia, ischemic stroke, transient ischemic attack (TIA), and seizures. However, similar to CJD, the presentation can be highly varied, and there is a reported case of PACNS that presented as a CJD-mimic, emphasizing the point that these two rare conditions can be difficult to differentiate [6].

Here, we report a case of neuropathology-confirmed CJD with an atypical presentation of progressive apraxia and aphasia, an unusually long disease duration of nearly five years, atypical multicentric plaques in the cerebellum, and a history of brain biopsy-proven PACNS 16 years prior. Interestingly, the location of brain biopsy was the left parietal lobe, the area localizing to her presenting CJD symptoms, raising the possibility of iatrogenic CJD.

## 2. Case History

A 58-year-old right-hand dominant woman presented to neurology in 2015, after two years of progressive neurological symptoms. Her initial symptom was clumsiness while writing, without overt tremor or micrographia. Over six months, this progressed to slowness of gait and dressing apraxia. After one year of symptoms, she developed word finding difficulty and memory loss. She also started to speak to photographs of people and mirror reflections, but without hallucinations per se. At the time of assessment, two years after onset, her comprehension was poor and her speech was non-fluent and perseverative, but she was fully mobile with no ataxia or myoclonus. There was some mood lability but no compulsions or behavioral changes, no fluctuations in levels of consciousness, no REM behavioral disorder, and no weight loss or rashes. She had had two falls when she turned to look behind her suddenly.

Pertinent past medical history included a seizure in 1996, a TIA in 1999, bilateral proximal leg weakness in 2011, and ischemic gut secondary to a transverse colon stricture in 2014. Her family history was negative for dementia, parkinsonism, amyotrophic lateral sclerosis (ALS), or mental illnesses. She was working as a school bus driver and chauffeur for several years prior to her first seizure. She had been a heavy drinker prior to 2009. She smoked for 28-pack-years.

The cause for her seizure, 17 years prior to her current symptom onset, had been investigated by CT, with discovery of a left posterior parietal mass that was surgically removed by craniotomy and pathologically proven to be PACNS (Figure 1C). No record is available as to what treatment she may have received then, other than Dilantin for several years, but she did not progress clinically at that time.

Two years prior to her current symptom onset, she was thought to have bilateral proximal leg weakness and mild atrophy, but EMG and multiple creatine kinase values were normal, so no muscle biopsy was pursued. The C-reactive protein (CRP) levels were grossly normal while she was symptomatic; there was an isolated elevated CRP (13.1) several months prior to her hemicolectomy, most likely secondary to ischemic colon, and this normalized (0.6) after the surgery.

On examination, she was alert but distracted, often pointing to and looking at the wall. Her speech was non-fluent and perseverative, and she was only able to follow some one-step commands. She was apraxic, unable to figure out how to hold a pen. A snout reflex was present, but no glabellar tap, grasp reflex, or rooting. Her cranial nerve assessment was unremarkable apart from impaired fixation. Eye movements were full with normal smooth pursuit. On motor examination, she had paratonia without spasticity or rigidity. There was generalized atrophy noted, but no evidence of fasciculations. Reflexes were 3+ in the upper limbs and 2+ in the lower limbs symmetrically. Toes were downgoing, and there was no clonus. She also had no ataxia on finger-nose testing or rapid alternating movements. Gait was slightly wide-based but not ataxic or bradykinetic. Arm swing was normal.

An MRI brain in the year prior to assessment had been reported as showing bilateral parietal atrophy, an electroencephalogram (EEG) showed diffuse slowing with no periodic discharges, and a lumbar puncture analysis was normal (14-3-3 not done and RT-QuIC was not yet available for diagnosis). However, closer inspection of the MRI revealed cortical ribboning in the right and left occipital cortices, left insula, and parietal, frontal, and cingulate cortices on both diffusion-weighted imaging (DWI) and fluid-attenuated inversion recovery (FLAIR) imaging (Figure 1A,B). A repeat EEG was performed and revealed generalized periodic sharp wave complexes consistent with CJD. Based on the presentation, MRI and EEG findings, a diagnosis of probable CJD was made.

Over the following years, her symptoms remained primarily aphasia and apraxia, such that she was able to remain mobile for several years, with evidence of ataxia and myoclonus only developing later, once she was wheelchair-bound. She died approximately five years after clinical onset.

In 2018, an autopsy of our patient revealed a globally atrophic brain (weight: 1070 g) with widespread and severe spongiform changes and a primarily synaptic pattern of prion protein staining (Figure 1E–H). No evidence of vessel inflammation was found. Interestingly, the molecular layer of the cerebellum contained plaques of prion protein, mostly in clusters, mimicking multicentric plaques, as often seen in the genetic prion disease GSS. No plaques were seen in other brain regions. Western blotting performed on the brain tissue showed a 2A pattern (Figure 1D) and genetic testing performed on brain tissue revealed no point mutations, deletions, or insertions of the PRNP gene. There was heterozygosity for the codon 129 locus (methionine-valine).

## 3. Discussion

Our patient was a 58-year-old woman with autopsy-proven CJD of the MV2 subtype, who presented with primary progressive apraxia and aphasia and survived for approximately five years, which is dramatically longer than the average CJD duration of 4–18 months. Her history included pathology-confirmed PACNS from a parietal resection in 1997, 16 years prior to the start of her symptoms. The original biopsy did not show signs of spongiform change, but it is interesting that the MRI done after symptom onset revealed more hyperintense signal near the area of the prior left parietal resection, which correlated with her initial symptoms of apraxia. This raises the question of whether an iatrogenic exposure during neurosurgery or PACNS itself had a role in the development of sCJD in our patient.

Iatrogenic cases of CJD have been linked to corneal transplant, stereotactic electroencephalogram electrodes, neurosurgical instruments, cadaveric dura mater, pituitary-derived growth hormone and blood transfusion from vCJD patients; incubation periods range from 1 to 42 years after exposure [3]. Overall, the incidence of proven iCJD from contaminated neurosurgical instrumentation is extremely low and all cases developed symptoms typical for sCJD (visual symptoms, dementia, and ataxia) within 2.3 years of exposure; three cases were identified in the 1950s [7] and one was reported in 1997 [8]. Case reports of CJD occurring up to 15 years after neurosurgery have also been reported [9], but without proof of a link to another case of CJD.

At least 228 cases of iCJD have occurred after exposure to cadaveric dura mater grafts, with incubation periods ranging from 16 months to 30 years [3]. While these patients also present clinically as sCJD, they can have kuru-like plaques on pathology, and this feature has been proposed as a means of identifying iCJD patients after exposure to contaminated dura mater graft [10].

Susceptibility to CJD is influenced by the polymorphism at codon 129 in the prion protein gene (PRNP); the methionine allele is overrepresented in sporadic and iatrogenic cases and heterozygosity is associated with longer incubation periods in iatrogenic cases [3]. The effect is not absolute, however, as MM homozygous individuals have had incubation periods longer than 30 years [3], and, in the 1997 case of neurosurgical transmission, the donor was MM homozygous and the recipient was MV, with an incubation period of only 26 months [8].

Turning to the potential role of inflammation in CJD, a literature search did not identify cases of CJD with a history of prior brain inflammation. However, there is evidence that chronic lymphocytic inflammation can enhance peripheral prion replication and may modify or influence iatrogenic transmission [11].

Our patient had multicentric plaques in the cerebellar molecular layer, which is atypical for sCJD. Such plaques are seen in genetic forms of CJD (GSS), but our patient had no *PRNP* mutations. Plaques can also be seen in iCJD, but they are usually of the kuru type and associated with the VV polymorphism, particularly in cases associated with dura mater graft contamination [10]. In sCJD of the MV2 subtype, as was our patient, kuru plaques in the cerebellum are also frequent [4], but the pattern of plaques in our case was clearly different. Our patient also had a prolonged course of almost five years, which is atypical for CJD of sporadic and iatrogenic cause, but there is a case report of a 56-year-old man who had a 42-month course of CJD, also with an MV2 subtype and presenting as primary progressive aphasia [12].

While it will be impossible to confirm whether our patient’s CJD was iatrogenic, it is striking that she would develop two rare conditions in the same area of the brain. The prior neurosurgical exposure to the parietal lobe could have been a source for the subsequent onset and spread of prion pathology, with the heterozygosity of codon 129 facilitating a prolonged incubation period. Her pathology did not have a “classic” iCJD phenotype, but it was also not typical for sCJD, given the multicentric plaques in the cerebellum. The long symptomatic period may have allowed more severe pathology to develop [13], but the plaque pattern seen in our patient was not described in the long duration MV2 case with primary progressive aphasia [12]. Finally, whether the inflammation from PACNS predisposed that area of the brain to prion propagation is another interesting speculation. In the end, our case highlights the variety of presenting symptoms and range of survival expectations in CJD, and the importance of asking about prior neurosurgical exposures and prior neuroinflammation. If other cases of non-genetic CJD are found that have multicentric PrP plaque deposits in the cerebellum, it may be worth considering iatrogenic exposures.

## Figures and Tables

**Figure 1 viruses-12-01411-f001:**
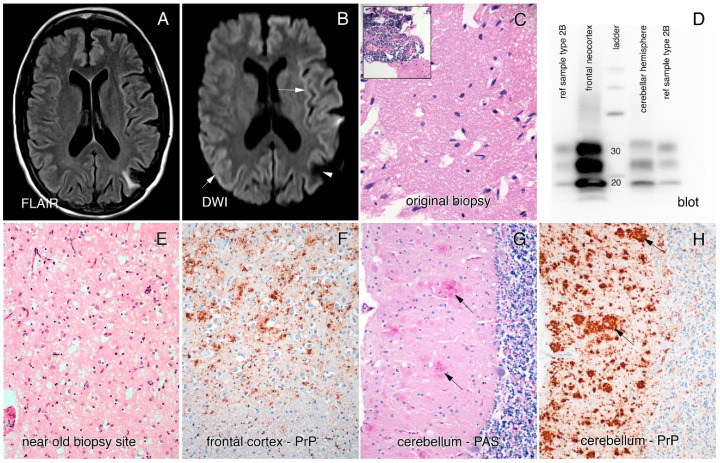
Brain MRI and pathology from original biopsy and final autopsy. (**A**,**B**) MRI brain. FLAIR (**A**) and DWI (**B**) showing area of previous biopsy in the left parietal lobe (white arrowhead) and hyperintense cortical ribboning with diffusion restriction (white arrows). (**C**) A 20× hematoxylin–eosin (H&E) section from the brain biopsy 20 years prior to death. It has several irregular clearings but lacks any true spongiform changes. The inset illustrates the different foci of perivascular (white arrow) lymphocytic infiltrates. (**D**) Immunoblot of proteinase K-digested PrP from frontal neocortex and cerebellar hemisphere, including two reference samples of type 2B CJD for comparison. The molecular weight of the lower band is consistent with type 2 and the glycoform pattern is type A. Prion antibody: 3F4. (**E**) A 20× H&E section of cortex near the old biopsy site from the autopsy, displaying abundant spongiform changes. (**F**) Similar magnification views from the frontal cortex, immunostained using the 12F10 anti-prion protein monoclonal antibody, which confirms the diagnosis of Creutzfeldt–Jakob disease. (**G**,**H**) cerebellum (PAS and 12F10, respective stains) demonstrating large multicentric-like plaques of prion protein (black arrows) without evidence for kuru plaques.

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
