# Peer review of "Creutzfeldt–Jakob Disease with a Five-Year Clinical Course, Multicentric Cerebellar Prion Plaques and Prior History of Biopsy-Proven Primary Angiitis of the Central Nervous System: A Case for Iatrogenic Exposure?"

_viruses, 2020, doi:10.3390/v12121411_

Round 1

Reviewer 1 Report

The paper describes one patient with atypical sCJD diagnosed with primary angiitis of the central nervous system by biopsy several years before. The post-mortem neuropathological study revealed spongiform encephalopathy with PrP-immunoreactive cluster-like (not really synaptic-like) aggregates in the cerebral cortex and PrP deposits consistent with confluent aggregates in the cerebellum. The patient had no mutations in the PRNP and a genetic and western blot study defined the pattern MV2 subtype. The authors discuss the possibility that the patient suffered from iatrogenic CJD. 

Some aspects deserve additional information: 1. characteristics of lesions in other regions of the brain; 2. regional microglial activation; 3. characteristics of proteinase K resistance of PrP; 4. identification of the region where the tissue was obtained to perform western blotting (it is workable that the band pattern of the cerebral cortex differs from that of the cerebellum); 5. identification of deposition of other proteins in the brain (i.e. P-tau); 6. definite absence of kuru plaques (correlation western blot and morphology in the cerebellum); 7. images of the western blots showing glycosylated, non-glycosylated bands, and absence of bands of very low molecular weight).

Author Response

We thank the reviewer for the thoughtful comments on our case report. The primary recommendation was for more pathology and strain-typing detail. We have done our best to address the seven suggestions as follows:

  1. characteristics of lesions in other regions of the brain;

We have included more pathology images from the cerebellum and frontal cortex for comparison (Figure 1 E-H) and have now specifically stated in the main text that plaques were not observed in any other brain regions (line 115).

  1. regional microglial activation;

The CJD Surveillance System in Canada does not routinely stain for microglia in cases of CJD, so we do not have any tissue stained for this. It is also unclear how the presence or pattern of microglial activation would change the underlying diagnosis of CJD. It also would not help us determine whether iatrogenesis is more likely, which is the intent of our case report, so we hope the reviewer will understand us not including microglial images.

  1. characteristics of proteinase K resistance of PrP;

We were able to contact the lab personnel at the National Microbiology Laboratory in Winnipeg who performed the PK digestion and immunoblot typing of PrP for this patient. The immunoblot showing pattern of PK digestion is now included in Figure 1, and the lab personnel are now included as co-authors.

  1. identification of the region where the tissue was obtained to perform western blotting (it is workable that the band pattern of the cerebral cortex differs from that of the cerebellum);

As mentioned above, we have now included immunoblot images from both frontal cortex and cerebellum in Figure 1, showing the same banding pattern.

  1. identification of deposition of other proteins in the brain (i.e. P-tau);

The CJD Surveillance System in Canada does not routinely stain for tau in cases of CJD, unless there is suspicion of a genetic case of GSS where there can be more tau pathology. We know this patient had no Prnp mutations so tau staining was not done. The plaques were also not exactly as seen in GSS, but more “multi-centric-like”. As we have no tissue stained for tau, we are not able to provide such images. Furthermore, the presence or absence of tau will not alter our diagnosis of CJD or help us determine an iatrogenic course, so we hope the reviewer will understand us not including tau images.

  1. definite absence of kuru plaques (correlation western blot and morphology in the cerebellum);

We have now provided western blot as well as imaging (PAS and PrP) of the plaques in the cerebellum to better demonstrate they are not kuru-like.

  1. images of the western blots showing glycosylated, non-glycosylated bands, and absence of bands of very low molecular weight).

Immunoblots of the samples have been included in Figure 2.

Overall, we feel that our manuscript has been greatly improved by the reviewer suggestions and that we have provided sufficient further information about the strain typing and pathology of this case to support our assertion that this is a case of definite CJD, MV2 subtype, that occurred 16 years after brain biopsy, with interesting clinical features, including a prolonged course, and interesting cerebellar pathology, but without proof of iatrogenesis. The intent of this as a clinical case report was to focus on the possible iatrogenic source of CJD and to give clinicians a reminder to consider such exposures if they see CJD patients with prior neurosurgery. In addition, we want our case to prompt others to publish similar case reports, to see if there is an unrecognized association between inflammatory brain conditions like PACNS and later onset of CJD, particularly if there is any overlap in the localization of symptom onset. Finally, in response to the other reviewer, we have also refocused our discussion on features of iatrogenesis, and contrasted our case findings with what is known about iatrogenic cases in the literature.

Reviewer 2 Report

This case report is interesting for the potential link between brain biopsy and the development of sCJD, for the particular long duration of the disease and for the pattern of PrP deposition.

First all, Creutzfeldt-Jakob should be written correctly! (see the title)

Second, the lag between brain biopsy and sCJD is sixteen years arguing on the potential iatrogenic etiology of sCJD and this is appropriately discussed.

However, this reviewer would suggest to be more focused on this issue avoiding to include in the discussion cases with PACNS mimicking CJD or single cases with a prolonged duration.  

Author Response

We thank the reviewer for providing feedback on our case report, including identifying the spelling error in the title! (how embarrassing…). We are glad that you found our report interesting and recognized its intent of raising the possibility of iatrogenic CJD. We also appreciate the suggestion of focusing the discussion more on iatrogenic cases, rather than PACNS mimicking CJD or single cases of prolonged duration. As such, we have largely rewritten the discussion, bringing in a more detailed look at the few neurosurgical transmission cases in the literature, the effect of 129 codon status on potential susceptibility to iatrogenesis and incubation period, and the potential influence of duration on plaque formation. This, in addition to the extra pathology and strain information we added to address the other reviewer comments, has made it a much more focused article that we hope will be informative and interesting.

Round 2

Reviewer 1 Report

The report is an interesting report which expands the coverage of prion diseases